# Burnout and Its Relationship with Work Engagement in Healthcare Professionals: A Latent Profile Analysis Approach

**DOI:** 10.3390/healthcare11233042

**Published:** 2023-11-26

**Authors:** David Luna, Rosa Paola Figuerola-Escoto, Juan José Luis Sienra-Monge, Alejandra Hernández-Roque, Arturo Soria-Magaña, Sandra Hernández-Corral, Filiberto Toledano-Toledano

**Affiliations:** 1Unidad de Investigación Multidisciplinaria en Salud, Instituto Nacional de Rehabilitación Luis Guillermo Ibarra Ibarra, Calzada México-Xochimilco 289, Arenal de Guadalupe, Tlalpan, Mexico City 14389, Mexico; xeurop@hotmail.com (D.L.); shcorral@gmail.com (S.H.-C.); 2Centro Interdisciplinario de Ciencias de la Salud Unidad Santo Tomás, Instituto Politécnico Nacional, Av. de Los Maestros s/n, Santo Tomás, Miguel Hidalgo, Mexico City 11340, Mexico; rfiguerolae@ipn.mx; 3Unidad de Pediatría Ambulatoria, Hospital Infantil de México Federico Gómez, National Institute of Health, Dr. Márquez 162, Doctores, Cuauhtémoc, Mexico City 06720, Mexico; jjsienra@hotmail.com (J.J.L.S.-M.); psicalehr@gmail.com (A.H.-R.); smarturo@yahoo.com.mx (A.S.-M.); 4Unidad de Investigación en Medicina Basada en Evidencias, Hospital Infantil de México Federico Gómez Instituto Nacional de Salud, Dr. Márquez 162, Doctores, Cuauhtémoc, Mexico City 06720, Mexico; 5Dirección de Investigación y Diseminación del Conocimiento, Instituto Nacional de Ciencias e Innovación para la Formación de Comunidad Científica, INDEHUS, Periférico Sur 4860, Arenal de Guadalupe, Tlalpan, Mexico City 14389, Mexico

**Keywords:** nurses, México, logistic regression, predictors, mental health, Spanish burnout inventory

## Abstract

The aim of this study was to use latent profile analysis to identify specific profiles of burnout syndrome in combination with work engagement and to identify whether job satisfaction, psychological well-being, and other sociodemographic and work variables affect the probability of presenting a profile of burnout syndrome and low work enthusiasm. A total of 355 healthcare professionals completed the Spanish Burnout Inventory, the Utrecht Work Engagement Scale, the Job Satisfaction Scale, and the Psychological Well-Being Scale for Adults. Latent profile analysis identified four profiles: (1) burnout with high indolence (BwHIn); (2) burnout with low indolence (BwLIn); (3) high engagement, low burnout (HeLb); and (4) in the process of burning out (IPB). Multivariate logistic regression showed that a second job in a government healthcare institution; a shift other than the morning shift; being divorced, separated or widowed; and workload are predictors of burnout profiles with respect to the HeLb profile. These data are useful for designing intervention strategies according to the needs and characteristics of each type of burnout profile.

## 1. Introduction

Burnout is defined as the result of chronic stress experienced in the work environment and can be contextualized within the framework of the coping model proposed by Lazarus and Folkman [1], which assumes the presence of cognitive and behavioral efforts developed to manage the external and/or internal demands that are in excess of the individual’s resources. Burnout was first described by Freudenberger [2], who highlighted a decrease in the energy of workers who were in direct contact with people; later, Maslach and Jackson [3] distinguished three dimensions of burnout in the work environment: (a) exhaustion, characterized by the worker’s experience of not being able to give more of himself at an affective level; (b) cynicism or depersonalization, which implies the development of negative attitudes and feelings toward the recipients of the work; and (c) reduced personal accomplishment or inefficacy, understood as the tendency of professionals to negatively evaluate their work activity and performance. From a similar point of view, the Eagly and Chaiken [4] model, that views attitudes as general evaluations in terms of favorability–unfavorability, influenced by affective experiences with the attitudinal object and cognitive components, was used by Gil-Monte [5] to explain the appearance of attitudes and characterized burnout as a response represented by cognitive deterioration (loss of enthusiasm toward the job) and emotional deterioration (psychological exhaustion) with the presence of negative attitudes (indolence), characterized by indifference or cold, distant and sometimes harmful behaviors toward the people to whom a service is provided. Gil-Monte [5] also observed that sometimes negative attitudes are accompanied by feelings of guilt, which represents a key variable in this model and can explain the relationship between burnout and variables such as depression [6].

The instruments for the evaluation of burnout were developed from the characterized models. The Maslach Burnout Inventory (MBI) [3,7] stands out in its various versions, being the most widely used instrument regardless of the characteristics of the sample. Several studies based on exploratory factor analysis (EFA) have reproduced its original structure (three factors) using orthogonal [8] or oblique [9] rotation or confirmatory factor analysis (CFA) [10,11,12]. But even with its proven usefulness, the MBI has some weaknesses, one of which is its factor structure, which denotes a theoretical foundation different from that offered in the manual [13], factorial ambiguity of some items [7,14], low reliability of the depersonalization scale [15], low discriminant validity with other related concepts [16] and the elaboration of different models derived from the different versions of the MBI [17]. For this reason, instruments that try to address these deficiencies have been developing, such as the Spanish Burnout Inventory (SBI), validated in the Spanish-speaking population (Argentina, Brazil, Chile, Colombia, Costa Rica, Mexico, Peru, Portugal, Uruguay and Spain). The SBI presents an internal consistency for the global score ranging from 0.76 to 0.90, which depends on the sex, country of origin and occupational sector of the sample, and an internal consistency for its individual factors ranging from 0.61 to 0.90, also based on the typical issues of the population in which it was calculated. The test–retest reliability coefficient was 0.76 for the global instrument and 0.62 to 0.76 for the individual factors. EFA identified a structure of four factors for this instrument (enthusiasm toward the job, psychological exhaustion, indolence and guilt) that explained 61.43% of the variance, and CFA determined that the detected model had an adequate fit that was reflected in the indices of X*, root mean square error of approximation (RMSEA), goodness-of-fit index (GFI), normed fit index (NFI) and comparative fit index (CFI), and it was concluded that the structural equation model showed a global fit to the observed data [18].

### 1.1. Burnout in Healthcare Professionals

Even though burnout can occur in any work context, its presence in healthcare professionals is striking, where it affects more than 50% of physicians [19,20] and at least 1 in 10 nurses worldwide [21]. Burden, long working hours, time pressure, role conflict and insecurity contribute to increasing the risk of burnout [22,23,24,25], and an association has also been found between this construct and variables such as routine, administrative work, regret over decisions regarding patients, managerial pressure and even workplace harassment or sacrifice of personal time [26]. Interestingly, single professionals (with or without a partner) were also found to be more likely to experience burnout [26], and physicians from racial and/or ethnic minorities experienced lower burnout rates [27].

The demonstrated consequences of burnout in healthcare personnel are anxiety, depression, insomnia, cognitive impairment, cardiovascular diseases and metabolic disorders [28,29,30,31,32]. Likewise, burnout has other negative consequences, such as lower work performance, worse results for patients, absenteeism, a low level of motivation, less job satisfaction, low efficiency and decreased productivity, which in turn can cause occupational disability and a poor quality of life [33,34,35,36]. There are also reports of disengagement at work and feelings of ineffectiveness [37,38,39]. Hence, the need to view burnout as a public health problem calls for both preventive and corrective interventions [40].

Multiple variables have been identified as protectors against burnout in healthcare professionals; one study points to the construct of moral resilience as a mitigant of exhaustion in health professionals. In general terms, the concept is based on a robust understanding of personal, relational and professional integrity and implies a focused and stable attention posture, precision about the values that are defended, nonreactive discernment to determine actions, coherence with one’s own values and, at times, the courageous act of speaking up and acting to defend them [41]. Lower rates of burnout are also found in people with a religious belief system [42] and those with social support and who practice self-compassion [43,44], which are seen as possible protective factors.

### 1.2. Work Engagement in Healthcare Professionals

Modifying the approach and focusing on positive psychology, which goes beyond the study of negative, dysfunctional or pathological aspects to the optimal functioning of people, teams and organizations [45], produces the engagement construct, also known as work enthusiasm, which is considered by some authors to be the opposite of burnout. Work engagement is defined as a positive mental state structured in three dimensions: vigor, dedication and absorption [46]. Vigor implies high levels of energy and mental resilience while working, a willingness to invest effort in the job and the persistence to continue despite difficulties or obstacles. Dedication refers to being strongly invested in the work and experiencing a sense of meaning along with strong inspiration, enthusiasm, pride and challenge. Finally, absorption implies a pleasant state characterized by total immersion or concentration in work so that time “flies by” [47]. Notably, a different and widely considered approach considers engagement as the opposite of burnout but within the same continuum, which is why it is structured from the positive equivalents of the dimensions in burnout (exhaustion, cynicism and inefficacy) [48]. In addition to its link with the concept of burnout, in the health context, associations have been detected between the level of engagement and the support of coworkers, rewards obtained, level of resilience, self-efficacy and optimism; it has also been negatively associated with cognitive demands [49]. Studies in the coronavirus disease 2019 (COVID-19) pandemic showed that at higher levels of distress in professionals, there are lower values of work engagement [50] and that a lower level of work engagement, in conjunction with a greater burden, conflicts and stressful situations, was related to a higher risk of psychological distress [51]. Other studies found that healthcare professionals who experience work enthusiasm manifest greater satisfaction and less workplace stress [52,53], which also impacts the way they treat the patient; “professionalism” increases empathy and improves work capacity [54], which results in a better quality of service provided to patients [55]. Finally, consistent with the theoretical framework, burnout has been placed as a negative predictor of work engagement, while life satisfaction and professional self-efficacy are positive predictors of work engagement, which suggests that professional self-efficacy, life satisfaction and burnout could impact job performance depending on work engagement [56]. These results are of great relevance, since traditionally, health personnel (as with other helping professions) have been one of the groups with the highest labor and emotional demands and with the highest levels of burnout (“being burned” in the work [57]).

As evidenced in this section, despite the emotional demands and psychosocial risks that their work imposes, the primacy of positive psychological effects manifested over negative ones in healthcare professionals has recently been found [58]. That is, these professionals have certain strengths, such as intrinsic motivation (for example, the vocation for their work), that help to increase their resources and their levels of work engagement [59]. Hence, it is important to evaluate variables that consider these strengths, such as work engagement or similar factors, such as job satisfaction, which is defined as a “positive or pleasant emotional state of the subjective perception of the subject’s work experiences” [58] and has been shown to be an indicator of well-being and quality of working life [59].

### 1.3. Person-Centered Approach in Burnout Research and Latent Profile Analysis (LPA)

Most of the approaches by which burnout has been studied are focused on variables [60,61,62,63] and are fundamentally aimed at finding associations of each of the dimensions of this construct separately with other variables in the studied sample. However, researchers are currently being encouraged to conduct person-centered studies on burnout issues. Research involving latent profile analysis (LPA) is derived from nonhierarchical cluster analysis procedures, such as the k-means method, and it is a relevant tool for placing objects in meaningful groups. In addition, this technique has advantages over unsupervised clustering techniques, such as the inclusion of stronger theoretical foundations, more clearly defined fit measures and the ability to perform confirmatory analyses [64].

In the person-centered approach, the sets of scores on scales and subscales are specific for each individual, which is different in a variable-centered approach, since in this, all the participants in the studied sample belong to a single population with a joint configuration of average parameters. In the person-centered approach, researchers account for the probability that the sample is made up of different subpopulations with different sets of parameters, which allows for the detection of differences between individuals, with which any phenomenon, such as burnout, can be studied from a different perspective [65]. This perspective recognizes the complexity of phenomena such as burnout, accounting for the system of variables that could be interrelated. These complex interactions could not be detected using a traditional or variable-centered approach [66].

Among previous studies that use a person-centered approach in burnout issues and LPA, Leiter and Maslach [67] performed an LPA to identify groups along the continuum from burnout to engagement in health professionals and found five profiles: (1) burnout, with high scores in all dimensions: cynicism, exhaustion and inefficacy; (2) engaged, with low scores in all dimensions: cynicism, exhaustion and inefficacy; (3) exhausted, with a high score in the exhaustion dimension and with moderate scores in cynicism and inefficacy; (4) unengaged, with a high score in the cynicism dimension and with moderate scores in exhaustion and inefficacy; and (5) ineffective, with a high score only in inefficacy and moderate scores in exhaustion and cynicism. In other words, they found three intermediate states along the continuum and gave greater importance to cynicism in the burnout experience. They also suggested that the prevalence of inefficacy implies the need for greater attention to this dimension in the burnout experience.

Another study that used LPA was carried out by Portoghese et al. [68] with Italian university students, in which they detected two profiles, one with high scores in the three dimensions of burnout (burned out) and another with low scores in the three dimensions (engaged), in addition to a profile of overextended people, which consisted of moderate scores in the cynicism and ineffectiveness dimensions and high scores in the exhaustion dimension, an issue that applied to 51% of the sample.

Another investigation with this analysis was carried out by Vinter [69] in adolescents in Estonia. In this study, two profiles were detected: students with “exhaustion above average” and students with “exhaustion below average”. “Above average” students showed lower buoyancy levels, and “below average” students used cognitive adaptive emotion regulation strategies less frequently (specifically the positive reappraisal strategy) and chose nonemotional strategies more often (especially blaming, reflecting and catastrophizing).

In Asia, Zhang et al. [70] identified four burnout profiles in a sample of middle school students in China: two of these profiles had high or low scores in all dimensions and two profiles had a mixture of scores, where we located a “persistent” group, with low scores in inefficacy and high scores in exhaustion and cynicism, and an opposite profile, with scores low on exhaustion and cynicism but high scores on inefficacy.

Lee et al. [71] considered two factors in a study of students in addition to burnout, cynicism and ineffectiveness: anxiety and resentment (antipathy). The authors found two groups with high scores and low scores on the five scales and identified two more profiles: one with low scores on inefficacy, exhaustion and anxiety and high scores on antipathy and cynicism, while the other group had high scores on inefficacy, exhaustion and anxiety and low scores on antipathy and cynicism.

In opposition to Maslach and Leiter [72], who consider burnout and engagement as opposite poles of the same dimension, Schaufeli and Salanova [73] reported that they are separate phenomena, on which the study by Salmela-Aro and Read [74] was based, involving students of higher education and resulting in four profiles: engaged, engaged–exhausted, inefficacious and burned out. The study concluded that as the years of study increased, burnout, cynicism and inefficacy increased. Another relevant result of this research was the detection of a large group of students who were burned and engaged at the same time (30%), which provides information about the unknown consequences of engagement.

Finally, a very recent study carried out with Polish students detected four profiles: low burnout (with low scores on the exhaustion and cynicism scales and moderate scores on the inefficacy scale), moderate but below average burnout (with low to moderate scores on the exhaustion and cynicism scales and moderate scores on the inefficacy scale), moderate but above average burnout (with moderate scores on all three scales) and burnout at high levels (with high or very high scores on the exhaustion and cynicism scales and moderate scores on the inefficacy scale) [75].

The results of the aforementioned studies denote that the phenomenon of burnout is complex and implies an individualized experience, and the distinction of specific profiles is required to develop intervention programs specific to the needs of the affected healthcare professionals.

### 1.4. Objectives of the Study

This study was inspired by the work of Maslach and Leiter [72], in which LPA was used to identify specific response patterns in the three dimensions of the MBI that indicate differentiated forms of behavior. However, it also differs from that study in three important aspects. First, according to Schaufeli and Bakker [76], burnout and work engagement are considered two opposite constructs that must be measured through specific instruments for each condition. Second, this study did not start with the five profiles identified by Maslach and Leiter [72] but determined the profiles from the LPA on the dimensions of the SBI. The SBI has the advantage of having been designed and validated in the cultural context of the target population and of having adequate psychometric properties for measuring burnout syndrome [77]. Finally, work engagement was measured using the Utrecht Work Engagement Scale (UWES-9), which made it possible to define specific profiles that combined both burnout syndrome and work engagement, maintaining the relative conceptual and behavioral independence of both constructs. Thus, the aim of this study was to use LPA to identify specific profiles of burnout syndrome in combination with work engagement. An additional objective was to estimate a multinomial logistic regression model to identify whether job satisfaction and psychological well-being, as well as other sociodemographic and work variables, affect the probability of presenting a profile denoting burnout syndrome and low work enthusiasm.

## 2. Materials and Methods

### 2.1. Participants

Using a nonprobabilistic convenience sampling technique, a sample of 355 health professionals was recruited. The subjects included 272 (76.6%) women and 83 (23.4%) men between 23 and 68 years old (M = 36.91; SD = 10.55); 215 (60.3%) subjects were single, 85 (23.9%) were married and 56 (15.8%) were separated, divorced or widowed. The subjects included medical (n = 232; 65.4%) and nursing (n = 123; 34.6%) personnel who worked either the morning shift (n = 176; 49.6%) or the evening, night, mixed or special guard shifts (n = 179; 50.4%), with employment seniority between 0 and 53 years (median = 3), who saw between 0 and 90 patients per day (median = 9). Only 31 (8.7%) subjects had a second job at another government healthcare institution, while 48 (13.5%) were in private practice. A total of 17.2% (n = 67) of subjects were diagnosed with a psychopathology in the last 12 months prior to the study, and 18.9% were prescribed psychotropic drugs in the same period. The inclusion criteria were personnel assigned to the Hospital Infantil de México Federico Gómez who agreed to voluntary participation in the study. The elimination criteria were not responding to 3 or more items in the same instrument or closing the window in which the online questionnaire was presented before completing it, which caused the responses not to be recorded.

### 2.2. Instruments

Except for the Sociodemographic Variables Questionnaire (Q-SV), the psychometric properties of the remaining instruments were verified in the sample by CFA using the weighted least squares means and variance adjusted (WLSMV) estimator with data from the polychoric correlation and asymptotic covariance matrix [78] from the reported structure for its validation in the Mexican population. The fit of each model was evaluated using the χ* test, χ****, RMSEA, weighted root mean square residual (WRMR), comparative fit index (CFI) and Tucker–Lewis index (TLI). Acceptable fit values were defined as follows: χ**** ≤ 5, RMSEA ≤ 0.08 (90% confidence interval (CI) < 0.10), WRMR ≤ 1, CFI ≥ 0.90, TLI ≥ 0.90. Excellent fit values were defined as follows: χ*****≤ 2, RMSEA ≤ 0.05, CFI ≥ 0.95, TLI ≥ 0.95 [79]. Finally, the Cronbach’s alpha of the instruments was estimated.

### 2.3. Sociodemographic Variables Questionnaire (Q-SV)

The Q-SV [80] was used to collect sociodemographic data (i.e., age, sex, marital status), as well as other labor variables, including medical or nursing personnel; seniority in years; morning shift or other shifts (i.e., evening, night, mixed or special guards: weekends, extended hours or on holidays); workload, defined as the number of patients seen daily; second job at another government healthcare institution or in private practice; and psychopathological diagnosis and/or prescription of psychoactive drugs in the 12 months prior to participating in this study.

### 2.4. Spanish Burnout Inventory (SBI)

The SBI [81,82] consists of 20 items answered on a Likert scale with 5 response options (i.e., 0: never to 4: very frequently or every day) organized into 4 dimensions: enthusiasm toward the job, psychological exhaustion, indolence and guilt. The score for each dimension is determined by the arithmetic sum of the scores for each item. The total score is calculated as the sum of the dimensions enthusiasm toward the job, psychological exhaustion and indolence divided by 15. The correction of this last score allows for its transformation into percentiles, and the burnout level was interpreted as very low (<11), low (11–33), medium (34–66), high (67–89) or critical (>89) [18]. The SBI theoretical model distinguishes between two types of burnout: guilt and nonguilt, whose difference lies in the presence or absence of guilt when detecting negative attitudes towards work. Nonguilt is defined as scores in the 4 dimensions equal to or greater than the 90th percentile. This instrument has been validated in Spanish with a population of various nationalities, including Mexican [18]. The psychometric properties of the confirmatory model with 4 factors were excellent (x*/df = 1.82; CFI = 0.99; TLI = 0.99; RMSEA = 0.04 (90% CI = 0.03 to 0.05)) except for x* (298.508, *p* < 0.01) and WRMR (1.05). The internal consistency for each dimension was α = 0.72, 0.87, 0.77 and 0.82 for enthusiasm toward the job, psychological exhaustion, indolence and guilt, respectively.

### 2.5. Utrecht Work Engagement Scale (UWES-9)

The validation of the UWES-9 [76] in the Mexican population was carried out in health professionals [83]. The scale comprises 9 items answered on a Likert scale with 6 response options (i.e., 0: never to 5: always) organized into three dimensions: vigor, dedication and absorption. The score for each dimension is estimated by the arithmetic sum of the scores for each item. The psychometric properties of the confirmatory model with 4 factors were excellent (x*/df = 1.99; CFI = 0.99; TLI = 0.99; RMSEA = 0.05 (90% CI = 0.03 to 0.07)) except for WRMR (0.798) and x2 (47.781, *p* < 0.01). However, the latter may be due to the sample size used in this study [84]. The internal consistency for each dimension was α = 0.70, 0.88 and 0.90 for the absorption, vigor and dedication dimensions, respectively.

### 2.6. Job Satisfaction Scale (JSS [85])

The JSS includes 15 items answered on a Likert scale with 7 response options (1: very dissatisfied to 7: very satisfied) organized into 2 dimensions: intrinsic motivation and extrinsic motivation. The version translated into Spanish by Tapia-Martínez et al. [86] and validated in the Mexican population of nursing personnel was used. Based on the Spanish validation carried out by Boluarte [87], two confirmatory structures were analyzed, one with a single factor and the other with two factors (i.e., original structure). The psychometric properties of the confirmatory model with 1 factor were excellent for the CFI (0.98) and TLI (0.97) but not acceptable for x* (493.232, *p* < 0.01), x*/df*(5.48), RMSEA (0.11 (90% CI = 0.10 to 0.13)) and WRMR (1.590). The internal consistency was α = 0.92.

### 2.7. Psychological Well-Being Scale for Adults (BIEPS-A [88])

The version of the BIEPS-A validated in the Mexican population [89] includes 9 items answered on a Likert scale with 3 response options (i.e., 1: disagree to 3: agree) organized into a single dimension. The total score is obtained by the arithmetic sum of the score for each item, and a score ≥26 is considered high psychological well-being. The psychometric properties of the confirmatory model with 1 factor were excellent (x*/df = 1.90; CFI = 0.99; TLI = 0.99; RMSEA = 0.05 (90% CI = 0.02 to 0.07); WRMR = 0.96) except for x* (51.446, *p* < 0.01). The internal consistency was α = 0.90.

### 2.8. Design

This was a cross-sectional and descriptive study. In addition to the variable-centered approach, a person-centered approach was also used. The first approach focuses on identifying the relationship between two or more variables in a sample taken as a homogeneous group of cases. On the other hand, the second focuses on detecting specific subgroups of cases in a sample that present defined and mutually exclusive response patterns, which implies a finer analysis of the behavior of the cases [90].

### 2.9. Procedure

Data collection was carried out by a researcher and a collaborator, both trained for this type of data collection, between September 2022 and January 2023 in a single session lasting approximately 10 min. The medical staff received a printed questionnaire, while the nursing staff received a link and a QR code to access the same questionnaire online, hosted on Google Forms*. The questionnaire included a cover letter that described the objective of the study, its benefits and the instructions to complete some instruments on their behaviors and attitudes during their work in the hospital; the email address of a researcher was provided as a means of contact to clarify doubts or answer any other requests. In addition, they were informed about the anonymity and confidentiality of their responses. At the end of the cover letter, a signature (medical staff) or checkbox with the phrase “I agree to participate in this study” (nursing staff) was requested from those who agreed to complete the instruments, which were then presented (c.f., [75]). A refusal to participate was not penalized in any way, and in this case, the return of the printed questionnaire was requested or they were asked to close the window in which the online version was presented. There is evidence that collecting information in print or online questionnaires is equivalent [91,92].

### 2.10. Ethical Considerations

This study is part of the research project “Salud mental positiva y su relación con la satisfacción y entusiasmo laboral y síndrome de quemarse por el trabajo durante la pandemia por COVID-19 en profesionales de la salud: un modelo predictivo” approved by the Committee of Research, Ethics in Research and Biosafety of the Hospital Infantil de México Federico Gómez (registration HIM-2021-054-FF). It is a study with minimal risk for the participants in accordance with the Regulations of the Ley General de Salud en Materia de Investigación para la Salud (Art. 3 Frac. I, Art. 4, Art. 6, Title II Chap. I, Art. 17 Frac. II) and its update published in the Diario Oficial de la Federación (April 2, 2014) and based on the Norma Oficial Mexicana NOM-012-SSA3-2012 (Section 5 numbers 5.3 to 5.13 and 5.15), which establishes the criteria for the execution of research projects for health in human beings. Its design and conduct were performed in accordance with national [93] and international [94] ethical guidelines for research in humans, as well as the Helsinki Declaration as updated in 2013 [95].

### 2.11. Data Analysis

The R v.4.3.1 program and its RStudio v.2023.06.1 interface were used for data analysis. The libraries employed were lavaan, psych, MVN, knitr, stats, PerformanceAnalytics, tidyLPA, rstatix, misty, lsr and nnet.

### 2.12. Burnout Prevalence

According to the interpretation of the SBI, the frequency and percentage of participants who presented different levels of burnout as well as the presence of guilt were estimated.

### 2.13. Descriptive Statistics and Correlation Analysis

The scores of each of the SBI dimensions (except guilt, since this is a categorical dimension of presence or absence) and the UWES-9 were analyzed. The mean, standard deviation and range were calculated, and their univariate and multivariate normalities were evaluated. For the latter, the Anderson–Darling test and the Mardia coefficient were used. Then, multivariate outliers were identified using the Mahalanobis distance with a cutoff point established by the χ2 value with degrees of freedom equal to the number of variables and *p* < 0.001. Multivariate outliers were retained for further analysis unless visual inspection of the data revealed a mechanistic pattern of responses [96]. Subsequently, the correlation between variables was estimated using Kendall’s τ coefficient. The strength of association was considered small, moderate and large at τ values ≥0.075, 0.225 and 0.375, respectively [97].

### 2.14. Latent Profile Analysis

The scores of each of the dimensions of the SBI (except guilt) and the UWES-9 were standardized to a mean of 0 and a variance of 1 (i.e., z scores), and an LPA was conducted with a model of variable means, equivalent variance between profiles and covariances fixed at 0 (EEI model [98]). To reach the study objectives, 3, 4 and 5 different profiles were estimated. The fit of the model for k profiles was evaluated using the Akaike information criterion (AIC), the consistent AIC (CAIC), the Bayesian information criterion (BIC), the sample size-adjusted Bayesian information criterion (SABIC), the bootstrap likelihood ratio test (BLRT-p) and entropy. The AIC, CAIC, BIC and SABIC criteria are indicators of model fit, and a lower value indicates a better fit. The BLRT indicates the maximum number of significant profiles in the model: a BLRT value for k profiles with *p* < 0.05 indicates a better fit with respect to k-1 profiles. The entropy acquires a value between 0 and 1, where a higher value indicates greater certainty in the classification of the observations in the corresponding profiles [99]. Together with these criteria, their parsimony, conceptual coherence and practical value were considered to determine the optimal number of profiles.

Once the optimal number of profiles was identified, the descriptive statistics for each variable in each profile were estimated with the real scores obtained. Prior to identifying differences between the variables in each profile, the Bartlett test was used to identify equal variances. When variances were equal, one-way ANOVA was performed; when they were not equal, one-way ANOVA was performed with Welch’s correction. In each case, the effect size was estimated using η2, which was interpreted as small, medium and large at η* ≥ 0.01, 0.06, 0.14 [100], respectively. For variables with significant differences (*p* ≤ 0.05), the Tukey test (one-way ANOVA) and the Games–Howell test (one-way ANOVA with Welch’s correction [101]) were performed.

### 2.15. Multinomial Logistic Regression

Once the k number of profiles was identified, multinomial logistic regression was performed to identify the predictors related to the probability of exhibiting the profile or profiles with the highest and lowest scores in the SBI and UWES-9 dimensions, respectively, regarding the profile with a reverse pattern of scores.

## 3. Results

### 3.1. Burnout Prevalence

Table 1 shows the frequencies and percentages of participants who presented different levels of burnout. Of the 14 participants with a critical level of burnout, 8 (57%) reported guilt.

### 3.2. Descriptive Statistics and Correlation Analysis

There was no evidence of univariate or multivariate normality (*p* < 0.01). Of a total of 355 observations, 7 (1.97%) were detected as multivariate outliers. Visual inspection of these observations did not reveal mechanical response patterns, so they were retained for all analyses. Table 2 shows the descriptive statistics for the SBI and UWES-9 dimensions and the Kendall τ correlations between these variables. Of the dimensions of the UWES-9 (i.e., absorption, vigor, dedication), enthusiasm toward the job (SBI) showed a positive correlation, while psychological exhaustion and indolence showed a negative correlation; the strength of the associations in all cases ranged from moderate to large.

### 3.3. Latent Profile Analysis

Table 3 shows the goodness-of-fit criteria, BLRT and entropy for the k profiles estimated in the model. From the values obtained in these indices and their parsimony for interpretation, the model with four profiles was retained (Figure 1). The profiles were classified as follows: (1) burnout with high indolence (BwHIn); (2) burnout with low indolence (BwLIn); (3) high engagement, low burnout (HeLb); and (4) in the process of burning out (IPB). Table 4 shows the descriptive statistics by profile for the variables evaluated.

The integration of these data revealed two profiles with high levels of burnout: BwHIn (n = 18; 5.1%) and BwLIn (n = 38; 10.7%). Both profiles showed low levels of enthusiasm toward the job (SBI) and vigor (UWES-9) and high levels of psychological exhaustion (SBI); however, the first profile showed a higher level of indolence (SBI) and lower absorption and dedication (UWES-9). The IPB profile (n = 100; 28.2%) was similar to the BwHIn profile in its high scores in psychological exhaustion and similar to the BwLIn profile in its indolence and absorption scores. Additionally, the three profiles with burnout each have participants who present guilt, BwHIn: 6; BwLIn: 1; IPB: 1. Finally, the HeLb profile (n = 199; 56.1%) showed the highest scores in each of the dimensions of the UWES-9 scale and the lowest in those of the SBI scale, which was a profile completely differentiated from the others.

### 3.4. Multinomial Logistic Regression

In the multinomial logistic regression, the response variable was the four identified profiles, and each profile was used as the reference category sequentially. The predictor variables were age, sex, marital status, position, shift, workload, seniority, second job in government and/or in private practice, level of psychological well-being and the intrinsic motivation and extrinsic motivation JSS scores. The odds ratios and the logistic regression coefficients for all predictors by each analyzed profile are shown in Table 5.

Compared to the HeLb profile, participants who work a shift other than the morning shift (OR = 5.71) and who have a second government job (OR = 13.25) are more likely to belong to the BwHIn profile; being divorced, widowed or separated (OR = 7.47) and having a higher workload (OR = 1.04) increased the probability of belonging to the BwLIn profile; and being divorced, widowed or separated (OR = 2.35) also increased the probability of belonging to the IPB profile.

Additionally, compared to the HeLb profile, older age (OR = 0.84), nurse position (OR = 0.10) and high psychological well-being (OR = 0.19) decreased the probability of belonging to the BwLIn profile, while the nurse position (OR = 0.37) decreased the probability of belonging to the IPB profile. Higher JSS scores decreased the probability of belonging to any of the profiles (BwHIn, OR = 0.91; BwLIn, OR = 0.89; IPB, OR = 0.95) compared to the HeLb profile.

## 4. Discussion

The main objective of this study was to use LPA to identify specific profiles of burnout syndrome in combination with work engagement. The results identified four nonredundant profiles with n ≥ 5% of the total sample, which are consistent with previous studies that have used LPA to identify specific burnout patterns (e.g., [102]) present in subgroups ≥ 1% of the sample evaluated [103]. Similar to other studies [67], two profiles with high levels of burnout were identified. In addition, these standard profiles extend the data from previous studies since they include the evaluation of work engagement as an independent, although related, concept to burnout [76]. The two standard profiles were the BwHIn profile and the HeLb profile. The first profile is characterized by a low level of absorption, vigor, dedication (UWES-9) and enthusiasm toward the job and a high level of psychological exhaustion and indolence (SBI). On the other hand, the second profile is characterized by a high level of absorption, dedication and enthusiasm toward the job and a low level of psychological exhaustion and indolence. The coherence and parsimony of these two profiles is reinforced by the data obtained in this study, which demonstrated a negative correlation between the three dimensions of the UWES-9 and the dimensions of psychological exhaustion and indolence of the SBI. At the same time, the dimension of enthusiasm toward the job of the SBI and the three dimensions of the UWES-9 exhibited a positive correlation. The evaluation and interpretation of work engagement independently of burnout is based on the conceptual distinction of both constructs made by Schaufeli and Bakker [76]. However, Leiter and Maslach [104] defend a position that implies a certain degree of independence and a certain degree of redundancy between both constructs. The present study partially reinforces both views. Supporting the redundancy of the two concepts, the pairs of dimensions, where each pair contains a member of each construct, enthusiasm toward the job/dedication and psychological exhaustion/vigor, showed a strong association regardless of the direction. This suggests opposite poles of a continuum, as defended by Leiter and Maslach [67], even when a different instrument than the one developed by these authors was used to measure burnout. The moderately strong associations in the remaining pairs of dimensions would support the independence of the two concepts. However, one of the intermediate profiles also supports this view, since it involved a particular combination of burnout and engagement that did not imply a linear relationship between them.

The LPA identified a second profile with burnout and low engagement (BwLIn), but it was qualitatively and quantitatively different from the first profile. This profile was not quantitatively different from the BwHIn profile regarding enthusiasm toward the job, psychological exhaustion and vigor, but it was different in terms of indolence, absorption and dedication. Additionally, the BwHIn profile had six of eight (75%) participants who felt guilty, while the BwLIn profile had only one of eight (12.5%) participants who felt guilty. Qualitatively, the BwLIn profile describes participants who, despite showing burnout syndrome, still show sensitivity to the problems or discomforts of the people they deal with, in this case, their patients. In addition, they show a higher level of absorption and dedication compared to those with the BwHIn profile, although they have the same level of vigor. This pattern indicates that they carry out activities that are meaningful to them with concern for the patient, but they do so with physical and emotional exhaustion and little personal satisfaction. This profile is relevant because it may imply the incorporation of other constructs that could provide relevant information to the study of burnout syndrome. It is about empathy and compassion fatigue. There is evidence that, regardless of their gender, people with greater empathic capacity prefer careers such as medicine and nursing [105], which plays a crucial role during their professional life, particularly in the healthcare professional–patient relationship [106]. However, a high level of empathy can lead to burnout syndrome and compassion fatigue [107]. This last concept refers to the physical and psychological disturbances that are caused by the empathic capacity of an individual that allows them to detect suffering in other people and has been reported in health professionals [108,109]. Thus, it is possible that the participants with the BwLIn profile have high levels of empathy that cause them compassion fatigue, which causes the highest burnout as well as the lowest engagement. However, this hypothesis still needs to be subjected to empirical evaluation.

The last profile identified was named “in the process of burning out” (IPB) due to its tendency toward a low level of enthusiasm toward the job, absorption, vigor and dedication and a tendency toward high levels of psychological exhaustion and indolence. This profile is quantitatively similar to the BwLIn profile in terms of indolence and absorption, so it is possible that it is a state prior to the BwLIn profile and even has the potential to develop into the BwHIn profile. Future studies should verify this hypothesis.

The total of four profiles is consistent with previous studies [110], although different from that reported in others (e.g., [111]). This suggests the complexity of the relationships between the components of burnout, as well as the fact that for this study, engagement was considered an independent concept that was measured with its own instrument and the scores were combined with those obtained in the SBI to establish the reported profiles.

The secondary objective was to estimate a multinomial logistic regression model to identify whether job satisfaction and psychological well-being, as well as other sociodemographic and occupational variables, affect the probability of presenting a profile denoting burnout syndrome and low work engagement. Compared with the probability of belonging to the HeLb profile, working the evening, night or mixed shift or having special guards or working a second job in another government institution increased the probability of belonging to the BwHIn profile by 5 to 13 times. Likewise, being divorced, separated or widowed increased the probability of belonging to the BwLIn profile by 7 times. Finally, being divorced, separated or widowed doubled the probability of belonging to the IPB profile. These results are consistent with previous studies that identified various occupational and sociodemographic factors as predictors of burnout in healthcare professionals (e.g., [112,113,114]). Unexpectedly, being a nursing staff member decreased the probability of belonging to the BwLIn and IPB profiles. This is counterintuitive because nursing staff are particularly vulnerable to stress given their level of empathy [115], which would make them more likely to present burnout. Job satisfaction as a factor that reduces the probability of belonging to a profile with burnout is also consistent with previous studies that identified various personal and psychological aspects of healthcare personnel as a protective factor against burnout [41,42,43,44]. In the case of psychological well-being, this can be interpreted as a salutogenic factor that decreases the probability of belonging at least to the BwLIn profile. This result is consistent with the literature indicating that the presence of salutogenic factors promotes positive mental health among healthcare personnel [19,20,22]. Older age was also a protective factor against belonging to this same profile, which is consistent with another study on burnout with a population similar to the one in this study [116] and may suggest the acquisition of skills to confront the work demands that lead to developing this syndrome. However, age was not significant in the remaining burnout profiles, so new research of this factor should be developed. Sex and marital status, particularly being married, were not factors related to burnout. This is consistent with some studies that indicate that sociodemographic factors are not predictors of burnout [117].

Finally, the percentage of participants with a critical level of burnout, which is defined as people with burnout syndrome according to the SBI, was 3.9%. These data are consistent with previous studies (e.g., [118]) and imply the need to design and implement preventive and intervention actions that can help alleviate this condition among healthcare professionals. In addition, the higher percentage of female participants is consistent with the feminization that has been reported for both careers in the healthcare area and its professionals [75]. This is relevant because it is the female population who is exposed to a greater number of conflicts between their professional life and their family life, which affect their mental health [118].

### 4.1. Practical Implications of this Work

The practical implications of this work are diverse. Both quantitative and qualitative differences between the two profiles with burnout syndrome (i.e., BwHIn and BwLIn) and the profile that is in the process (i.e., IPB) indicate that each of them requires specific attention according to their particularities. Additionally, in this study it was detected that some of the variables that predict belonging to one or another profile are difficult to manipulate (e.g., BwHIn: shift; BwLIn: nurse) because they are constrained by socioeconomic factors or refer to working conditions present in the care institution where healthcare professionals work. The above implies that the strategies designed should focus rather on psychological variables that allow healthcare professionals to functionally face the demands to which they are exposed. In this regard, there is evidence that moral resilience [41], resilience and social support [43] and self-compassion [44] are factors that prevent deterioration in the mental health of this population, including the presence of burnout syndrome. However, the psychological variables that affect each of these two profiles and the way in which they do so remain unknown. It is likely that they act in a differentiated manner, which would suggest the need for intervention strategies specific to each profile, instead of generic interventions. The identification of the IPB profile also implies the need for focused attention to its needs, which prevents its members from evolving to either the BwHIn or BwLIn profile. Finally, further investigation of the HeLb profile may be useful to obtain information about the psychological characteristics of healthcare professionals who, although exposed to the same demands as their colleagues with burnout, do not develop this syndrome. The latter could be used in burnout syndrome prevention programs that instruct or enhance these characteristics.

An unexpected aspect that could have practical implications in burnout research is the fact that divorced, separated or widowed people, although they present burnout, are not found in the BwHIn profile. One possibility is that, by not having a partner, they avoid experiencing marital burnout [119], which can serve as an additional factor to the work demands to which they are exposed and that can enhance the development of burnout. Future studies should investigate the role of marital burnout in healthcare professionals in order to identify its role in their work burnout.

### 4.2. Limitations

This study has several limitations that affect the external validity of its results. The first is that healthcare professionals from a single care center in a specific region of the country were analyzed. The second is that a nonrandom sample of a reduced size was used. Future studies should increase the size of the sample, preferably selected using a random technique, as well as increase the number of healthcare centers in various geographical areas of the country. Third, despite the use of innovative data analysis techniques, such as LPA, causal relationships were not evaluated in this study. This deficiency can be treated in two different ways in future studies. On the one hand, the use of structural equation models would allow the evaluation of hypothetical causal relationships between the variables evaluated and the interpretations made. A specific case would be to evaluate the role of empathy and compassion fatigue in the development of burnout syndrome. On the other hand, based on the identified profiles, it would be possible to design intervention strategies and evaluate their effectiveness through controlled clinical trials. Finally, other psychosocial variables should be considered in their relationship with burnout, specifically those related to salutogenic factors. This would be important to identify possible protective factors with cognitive and behavioral aspects that could be trained in healthcare professionals so that they would have a greater repertoire of skills that would allow them to functionally face the demands to which they are exposed and thus reduce the probability of developing burnout syndrome.

## 5. Conclusions

Through LPA, two profiles were identified in healthcare professionals that combined burnout syndrome and low work enthusiasm in a specific and mutually exclusive way. A third profile exhibited an inverse pattern, with a low level of burnout and high work engagement. The last profile included people who tended to develop burnout syndrome and low work engagement. The differences between the two profiles with burnout suggest the participation of empathy and compassion fatigue, possibly as mediating variables between burnout and work engagement. This relationship should be investigated in future studies. It is also necessary to design and implement strategies that reverse the tendency to burnout and low work enthusiasm detected as a specific profile. A second job in a government healthcare institution; a shift other than the morning shift; being divorced, separated or widowed; and the workload are predictors of burnout syndrome and low work engagement. Positive psychological variables, such as well-being, that prevent burnout should be investigated.

## Figures and Tables

**Figure 1 healthcare-11-03042-f001:**
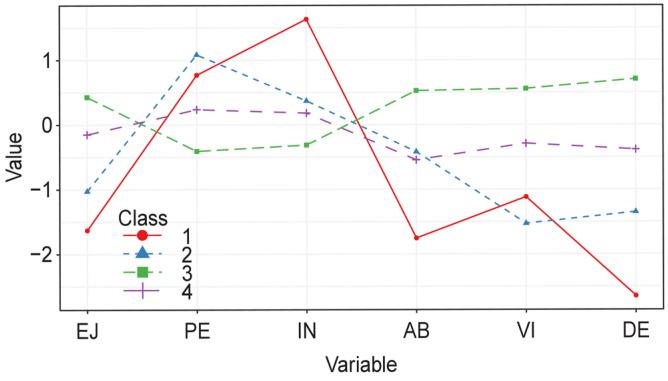
Analysis of latent profiles for the scores in the dimensions of the Spanish Burnout Inventory and the Utrecht Work Engagement Scale. EJ: enthusiasm toward the job; PE: psychological exhaustion; IN: indolence; AB: absorption; VI: vigor; DE: dedication; Class 1: burnout with high indolence; Class 2: burnout with low indolence; Class 3: high engagement, low burnout; Class 4: in the process of burning out.

**Table 1 healthcare-11-03042-t001:** Prevalence of burnout syndrome in healthcare professionals.

Variables	Very Low	Low	Medium	High	Critical
Frequency	66	101	132	42	14
Percentage	18.6	28.5	37.2	11.8	3.9

**Table 2 healthcare-11-03042-t002:** Descriptive statistics for the dimensions of the Spanish Burnout Inventory and the Utrecht Work Engagement Scale and their correlation.

Variable	M (SD)	Range (Min–Max)	Psychological Exhaustion	Indolence	Absorption	Vigor	Dedication
Enthusiasm toward the job	17.65 (2.65)	6–20	−0.26 *	−0.21 *	0.29 *	0.36 *	0.47 *
Psychological exhaustion	7.18 (4.02)	0–16	1	0.26 *	−0.16 *	−0.45 *	−0.36 *
Indolence	3.56 (3.34)	0–21		1	−0.21 *	−0.23 *	−0.28 *
Absorption	12.28 (2.05)	1–15			1	0.44 *	0.49 *
Vigor	11.33 (2.71)	0–15				1	0.55 *
Dedication	13.53 (1.80)	5–15					1

Abbreviations: M: mean; SD: standard deviation; Min: minimum value; Max: maximum value. Notes: * *p* < 0.001.

**Table 3 healthcare-11-03042-t003:** Goodness-of-fit criteria, the bootstrap likelihood ratio test and the entropy for k estimated profiles.

Profiles	AIC	CAIC	BIC	SABIC	BLRT-p	Entropy
Indolence	5353.50	5480.17	5454.18	5371.69	0.01	0.91
Absorption	5272.96	5433.74	5400.74	5296.05	0.01	0.90
Vigor	5236	5430.88	5390.88	5263.98	0.01	0.85

Abbreviations: AIC: Akaike information criterion; CAIC: consistent AIC; BIC: Bayesian information criterion; SABIC: sample size-adjusted Bayesian information criterion; BLRT-p: bootstrap likelihood ratio test *p*-value.

**Table 4 healthcare-11-03042-t004:** Descriptive statistics for the scores in the dimensions of the Spanish burnout inventory and the Utrecht work engagement scale for each profile.

Variables	M (SD)				f	η *	p.h.
	Burnout with High Indolence	Burnout with Low Indolence	High Engagement, Low Burnout	In Progress of Burning Out			
EJ	13.33 (2.80)	14.84 (2.12)	18.78 (1.87)	17.22 (2.41)	58.22 ***	0.37	1 and 2 ≠ 3 ≠ 4
PE *	10.27 (3.92)	11.68 (3.01)	5.57 (3.34)	8.12 (3.72)	43.26 ***	0.26	1 ≠ 3, 2 ≠ 3 ≠ 4
IN	9.05 (5.09)	4.57 (3.22)	2.52 (2.47)	4.24 (3.29)	18.04 ***	0.21	2 and 4 ≠ 1 ≠ 3
AB	8.77 (2.64)	11.39 (1.68)	13.35 (1.38)	11.12 (1.62)	64.82 ***	0.42	2 and 4 ≠ 1 ≠ 3
VI	8.33 (2.74)	7.05 (2.40)	12.79 (1.77)	10.57 (1.69)	94.31 ***	0.52	1 and 2 ≠ 3 ≠ 4
DE	8.77 (1.30)	11.10 (0.72)	14.79 (0.42)	12.78 (0.75)	558.74 ***	0.88	1 ≠ 2 ≠ 3 ≠ 4

Abbreviations: AB: absorption; DE: dedication; EJ: enthusiasm toward the job; IN: indolence; p.h.: post hoc; PE: psychological exhaustion; VI: vigor. Notes: *** *p* < 0.01. * *p* < 0.001.

**Table 5 healthcare-11-03042-t005:** Multinomial logistic regression coefficients for latent profile analysis, with HeLb as reference.

Variables	BwHIn				BwLIn				IPB			
	B	SE	p	OR	B	SE	p	OR	B	SE	p	OR
Age	−0.01	0.07	0.87	0.98	−0.16	0.06	0.01	0.84	−0.02	0.03	0.37	0.97
Sex (female)	−0.19	0.66	0.76	0.82	−0.18	0.55	0.73	0.82	−0.17	0.33	0.60	0.84
CS (married)	0.04	0.89	0.90	1.04	0.78	0.71	0.26	2.19	0.05	0.38	0.88	1.05
CS (other)	−0.50	1.41	0.71	0.60	2.01	0.79	0.01	7.47	0.85	0.44	0.05	2.35
Pos (nurse)	−1.24	1.10	0.25	0.28	−2.29	0.88	<0.01	0.10	−0.98	0.40	0.01	0.37
Shift (other)	1.74	0.85	0.04	5.71	−0.72	0.54	0.18	0.48	−0.13	0.40	0.66	0.87
Wl	−0.01	0.03	0.63	0.98	0.03	0.02	0.05	1.04	0	0.01	0.91	0.99
Senior	−0.02	0.08	0.74	0.97	0.03	0.05	0.51	1.03	0	0.02	0.80	0.99
2GJ (yes)	2.58	1.01	0.01	13.25	−0.92	1.13	0.41	0.39	0.56	0.48	0.24	1.75
2PP (yes)	−1.45	1.29	0.26	0.23	−0.42	0.81	0.60	0.65	0.08	0.43	0.84	1.09
JSS	−0.08	0.01	<0.01	0.91	−0.11	0.01	<0.01	0.89	−0.05	0	<0.01	0.95
BIEPS-A (high)	−0.75	0.61	0.22	0.46	−1.64	0.50	<0.01	0.19	−0.37	0.30	0.22	0.68
PsPDx (yes)	0.77	0.88	0.30	2.16	0.14	0.72	0.84	1.15	−0.19	0.47	0.67	0.82
PresPsyD (yes)	−0.01	0.88	0.98	0.98	0.46	0.69	0.50	1.59	−0.14	0.45	0.74	0.86

Abbreviations: 2GJ: second government job; 2PP: second job in private practice; BIEPS-A: psychological well-being; BwHIn: burnout with high indolence; BwLIn: burnout with low indolence; CS: civil status; HeLb: high engagement, low burnout; IPB: in process of burning out; JSS: job satisfaction; Pos: position; PresPsyD: prescription of psychotropic drugs; PsPDx: psychopathology diagnosis; Senior: seniority; Wl: number of patients attended in a shift.

## Data Availability

The raw data supporting the conclusions of this article will be made available by the authors under a reasonable request.

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
