# Peer review of "Burnout and Its Relationship with Work Engagement in Healthcare Professionals: A Latent Profile Analysis Approach"

_healthcare, 2023, doi:10.3390/healthcare11233042_

Round 1

Reviewer 1 Report

The manuscript "Burnout and its Relationship with Work Engagement in Healthcare Professionals: A Latent Profile Analysis Approach" is a very well written and interesting article.  I have attached a Word file with comments provided on the manuscript that reflect minor points of clarification that should be considered.  The only major comment is that I was disappointed with the brevity of and general lack of insight reflected in Section 4.1 - Practical Implications of the Work.  Because of the interesting results of this study, I believe additional insights could be provided by the authors on the practical implications and possible interventions they see being a possible outgrowth of their research results.  Otherwise, it is an excellent manuscript. 

Author Response

Review Report (Reviewer 1)

Comments:

I would reword the sentence as: "The demonstrated consequences of burnout in healthcare personnel are anxiety, depression...

Response:

The comment was addressed.

Change:

The demonstrated consequences of burnout in healthcare personnel are anxiety, depression, insomnia, cognitive impairment, cardiovascular diseases and metabolic disorders [28–32]. Likewise, burnout has other negative consequences, such as lower work performance, worse results for patients, absenteeism, a low level of motivation, less job satisfaction, low efficiency and decreased productivity, which in turn can cause occupational disability and a poor quality of life [33–36].

Comments:

Please place a comma after "aspects"

Response:

The comment was addressed.

Change:

Modifying the approach and focusing on positive psychology, which goes beyond the study of negative, dysfunctional or pathological aspects, to the optimal functioning of people, teams and organizations [45], produces the engagement construct, also known as work enthusiasm, which is considered by some authors to be the contrary of burnout.

Comments:

You probably need to provide a few more sentences on describing what LPA is to your audience beyond stating it is a means to classify objects into meaningful groups.

Response:

The comment was addressed.

Change:

Most of the approaches by which burnout has been studied are focused on variables [60–63] and are fundamentally aimed at finding associations of each of the dimensions of this construct separately with other variables in the studied sample. However, researchers are currently being encouraged to conduct person-centered studies on burnout issues. Research involving latent profile analysis (LPA) is derived from nonhierarchical cluster analysis procedures, such as the k-means method, and is a relevant tool for placing objects in meaningful groups, in addition this technique has advantages over unsupervised clustering techniques, such as the inclusion of stronger theoretical foundations, more clearly defined fit measures, and the ability to perform confirmatory analyzes [64].

Comments:

Not sure what "closing an online battery" means in this context. Do you mean losing power? Or closing the survey before completing it?

Response:

The comment was addressed.

Change:

2.1. Participants

Using a nonprobabilistic convenience sampling technique, a sample of 355 health professionals was recruited. The subjects included 272 (76.6%) women and 83 (23.4%) men between 23 and 68 years old (M = 36.91; SD = 10.55); 215 (60.3%) subjects were single, 85 (23.9%) were married and 56 (15.8%) were separated, divorced or widowed. The subjects included medical (n = 232; 65.4%) and nursing (n = 123; 34.6%) personnel who worked either the morning shift (n = 176; 49.6%) or the evening, night, mixed or special guard shifts (n = 179; 50.4%), with employment seniority between 0 and 53 years (median = 3), who saw between 0 and 90 patients per day (median = 9). Only 31 (8.7%) subjects had a second job at another government health care institution, while 48 (13.5%) were in private practice. A total of 17.2% (n = 67) of subjects were diagnosed with some psychopathology in the last 12 months prior to the study, and 18.9% were prescribed psychotropic drugs in the same period. The inclusion criteria were personnel assigned to the Hospital Infantil de México Federico Gómez who agreed to voluntary participation in the study. The elimination criteria were not responding to 3 or more items in the same instrument or close the window in which the online questionnaire was presented before completing it, which caused the responses not to be recorded.

Comments:

Not sure what you mean by "special guards" in terms of shiftwork. Need to clarify perhaps.

Response:

The comment was addressed.

Change:

2.3. Sociodemographic Variables Questionnaire (Q-SV)

The Q-SV [80] was used to collect sociodemographic data (i.e., age, sex, marital status), as well as other labor variables, including medical or nursing personnel; seniority in years; morning shift or other shifts (i.e., evening, night, mixed or special guards: weekends, extended hours or on holidays); workload, defined as the number of patients seen daily; second job at another government health care institution or in private practice; and psychopathological diagnosis and/or prescription of psychoactive drugs in the 12 months prior to participating in this study.

Comments:

It is not obvious why you chose these percentages as being representative of the descriptive levels presented. Perhaps adding a sentence or two could explain your logic. I would imagine having an even distribution of the five categories with equal percentage ranges (very low - less than 21), low - 21 to 40, medium 41 - 60, high - 61 - 80, critical, 81 to 100 percent. Without explanation or reference, your percentage categories seem arbitrary and capricious.

Response:

The comment was addressed.

Change:

2.4. Spanish Burnout Inventory (SBI)

The SBI [81,82] consists of 20 items answered on a Likert scale with 5 response options (i.e., 0: never to 4: very frequently or every day) organized into 4 dimensions: enthusiasm toward the job, psychological exhaustion, indolence, and guilt. The score for each dimension is determined by the arithmetic sum of the scores for each item. The total score is calculated as the sum of the dimensions enthusiasm toward the job, psychological exhaustion and indolence divided by 15. The correction of this last score allows for its transformation into percentiles, and the burnout level was interpreted as very low (<11), low (11-33), medium (34-66), high (67-89), or critical (>89) [18]. The SBI theoretical model distinguishes between two types of burnout: guilt and nonguilt. Nonguilt is defined as scores in the 4 dimensions equal to or greater than the 90th percentile. This instrument has been validated in Spanish with a population of various nationalities, including Mexican [18]. The psychometric properties of the confirmatory model with 4 factors were excellent (x2/df = 1.82; CFI = 0.99; TLI = 0.99; RMSEA = 0.04 (90% CI = 0.03 to 0.05)) except for x2 (298.508, p < 0.01) and WRMR (1.05). The internal consistency for each dimension was α = 0.72, 0.87, 0.77, and 0.82 for enthusiasm toward the job, psychological exhaustion, indolence, and guilt, respectively.

Comments:

If your sample has a large n in CFA, then having a low p value for x2 is a common bias observed often. I am sure there is a reference on this somewhere.

Response:

The comment was addressed.

Change:

The validation of the UWES-9 [76] in the Mexican population was carried out in health professionals [83]. The scale comprises 9 items answered on a Likert scale with 6 response options (i.e., 0: never to 5: always) organized into three dimensions: vigor, dedication, and absorption. The score for each dimension is estimated by the arithmetic sum of the scores for each item. The psychometric properties of the confirmatory model with 4 factors were excellent (x2/df = 1.99; CFI = 0.99; TLI = 0.99; RMSEA = 0.05 (90% CI = 0.03 to 0.07)) except for WRMR (0.798) and x2 (47.781, p < 0.01). However, the latter may be due to the sample size used in this study (Kyriazos, 2018). The internal consistency for each dimension was α = 0.70, 0.88, and 0.90 for the absorption, vigor and dedication dimensions, respectively.

Comments:

I would reword: "Of the dimensions of the UWES-9 (absorption, vigor and dedication), enthusiasm…"

Response:

The comment was addressed.

Change:

3.2. Descriptive Statistics and Correlation Analysis

There was no evidence of univariate or multivariate normality (p < 0.01). Of a total of 355 observations, 7 (1.97%) were detected as multivariate outliers. Visual inspection of these observations did not reveal mechanical response patterns, so they were retained for all analyses. Table 2 shows the descriptive statistics for the SBI and UWES-9 dimensions and the Kendall τ correlations between these variables. Of the dimensions of the UWES-9 (i.e., absorption, vigor, dedication), enthusiasm toward the job (SBI) showed a positive correlation, while psychological exhaustion and indolence showed a negative correlation; the strength of the associations in all cases ranged from moderate to large.

Comments:

In progress of burning out

Response:

The comment was addressed.

Change:

Variables

M (SD)

f

η2

p.h.

Burnout with high indolence

Burnout with low indolence

High engagement, low burn

In progress of burning out

EJ

13.33 (2.80)

14.84 (2.12)

18.78 (1.87)

17.22 (2.41)

58.22***

0.37

1 and 2 ≠ 3 ≠ 4

PE

10.27 (3.92)

11.68 (3.01)

5.57 (3.34)

8.12 (3.72)

43.26***

0.26

1 ≠ 3, 2 ≠ 3 ≠ 4

IN

9.05 (5.09)

4.57 (3.22)

2.52 (2.47)

4.24 (3.29)

18.04***

0.21

2 and 4 ≠ 1 ≠ 3

AB

8.77 (2.64)

11.39 (1.68)

13.35 (1.38)

11.12 (1.62)

64.82***

0.42

2 and 4 ≠ 1 ≠ 3

VI

8.33 (2.74)

7.05 (2.40)

12.79 (1.77)

10.57 (1.69)

94.31***

0.52

1 and 2 ≠ 3 ≠ 4

DE

8.77 (1.30)

11.10 (0.72)

14.79 (0.42)

12.78 (0.75)

558.74***

0.88

1 ≠ 2 ≠ 3 ≠ 4

Comments:

Not sure I understand this sentence - what 3 profiles are you referring to? And what does 6,1,1 refer to? You may need to clarify.

Response:

The comment was addressed.

Change:

The integration of these data revealed two profiles with high levels of burnout: BwHIn (n = 18; 5.1%) and BwLIn (n = 38; 10.7%). Both profiles showed low levels of enthusiasm toward the job (SBI) and vigor (UWES-9) and high levels of psychological exhaustion (SBI); however, the first profile showed a higher level of indolence (SBI) and lower absorption and dedication (UWES-9). The IPB profile (n = 100; 28.2%) was similar to the BwHIn profile in its high scores in psychological exhaustion and similar to the BwLIn profile in its indolence and absorption scores. Additionally, the three profiles with burnout each have participants who present guilt, BwHIn: 6; BwLIn: 1; IPB: 1. Finally, the HeLb profile (n = 199; 56.1%) showed the highest scores in each of the dimensions of the UWES-9 scale and the lowest in those of the SBI scale, which was a profile completely differentiated from the others.

Comments:

In the legend be consistent with B and p - p should not be capitalized; B probably should be capitalized, but whatever you choose be consistent

Response:

The comment was addressed.

Change:

Variables

BwHIn

BwLIn

IPB

B

SE

p

OR

B

SE

p

OR

B

SE

p

OR

Age

−0.01

0.07

0.87

0.98

−0.16

0.06

0.01

0.84

−0.02

0.03

0.37

0.97

Sex (female)

−0.19

0.66

0.76

0.82

−0.18

0.55

0.73

0.82

−0.17

0.33

0.60

0.84

CS (married)

0.04

0.89

0.90

1.04

0.78

0.71

0.26

2.19

0.05

0.38

0.88

1.05

CS (other)

−0.50

1.41

0.71

0.60

2.01

0.79

0.01

7.47

0.85

0.44

0.05

2.35

Pos (nurse)

−1.24

1.10

0.25

0.28

−2.29

0.88

< 0.01

0.10

−0.98

0.40

0.01

0.37

Shift (other)

1.74

0.85

0.04

5.71

−0.72

0.54

0.18

0.48

−0.13

0.40

0.66

0.87

Wl

−0.01

0.03

0.63

0.98

0.03

0.02

0.05

1.04

0

0.01

0.91

0.99

Senior

−0.02

0.08

0.74

0.97

0.03

0.05

0.51

1.03

0

0.02

0.80

0.99

2GJ (yes)

2.58

1.01

0.01

13.25

−0.92

1.13

0.41

0.39

0.56

0.48

0.24

1.75

2PP (yes)

−1.45

1.29

0.26

0.23

−0.42

0.81

0.60

0.65

0.08

0.43

0.84

1.09

JSS

−0.08

0.01

< 0.01

0.91

−0.11

0.01

< 0.01

0.89

−0.05

0

< 0.01

0.95

BIEPS−A (high)

−0.75

0.61

0.22

0.46

−1.64

0.50

< 0.01

0.19

−0.37

0.30

0.22

0.68

PsPDx (yes)

0.77

0.88

0.30

2.16

0.14

0.72

0.84

1.15

−0.19

0.47

0.67

0.82

PresPsyD (yes)

−0.01

0.88

0.98

0.98

0.46

0.69

0.50

1.59

−0.14

0.45

0.74

0.86

Comments:

How do you explain that being divorced, widowed or separated affected the BwLIn and IPB profiles, and not the the BwHIn profile?

Response:

The comment was addressed.

Change:

An unexpected aspect that could have practical implications in burnout research is the fact that divorced, separated or widowed people, although they present burnout, are not found in the BwHIn profile. One possibility is that, by not having a partner, they avoid experiencing marital burnout [Nejatian et al. 2021], which can serve as an additional factor to the work demands to which they are exposed and that can enhance the development of burnout. Future studies should investigate the role of marital burnout in healthcare professionals in order to identify its role in their work burnout.

Comments:

What do you mean by "standard"?

Response:

It was a mistake to point out the profiles as standard. This word was eliminated, without conceptual or practical consequences for this study.

Change:

The main objective of this study was to use LPA to identify specific profiles of burnout syndrome in combination with work engagement. The results identified 4 nonredundant profiles with n ≥ 5% of the total sample, which are consistent with previous studies that have used LPA to identify specific burnout patterns (e.g., [101]) present in subgroups ≥ 1% of the sample evaluated [102]. Similar to other studies [67], two profiles with high levels of burnout were identified. In addition, these standard profiles extend the data from previous studies since they include the evaluation of work engagement as an independent, although related, concept to burnout [76]. The two standard profiles were the BwHIn profile and the HeLb profile. The first profile is characterized by a low level of absorption, vigor, dedication (UWES-9) and enthusiasm toward the job and a high level of psychological exhaustion and indolence (SBI). On the other hand, the second profile is characterized by a high level of absorption, dedication and enthusiasm toward the job and a low level of psychological exhaustion and indolence.

Comments:

This "0.04" times result is hardly worth mentioning compared to other higher ORs.

Response:

Given its practical irrelevance, the low oddratio was eliminated from this part of the discussion.

Change:

Major comments:

Table 2: Also report the observed frequencies of the four item categories in the table. In addition, include the polychoric correlation matrix of all 39 items in a supplementary material or make your dataset publicly available.

Response:

The additional material presents the matrix of polychoric correlations which were estimated with the FACTOR V. 12.04.04 (WIN64) program.

Change:

The secondary objective was to estimate a multinomial logistic regression model to identify whether job satisfaction and psychological well-being, as well as other sociodemographic and occupational variables, affect the probability of presenting a profile denoting burnout syndrome and low work engagement. Compared with the probability of belonging to the HeLb profile, working the evening, night, or mixed shift or having special guards or working a second job in another government institution increased the probability of belonging to the BwHIn profile by 5 to 13 times. Likewise, being divorced, separated or widowed increased the probability of belonging to the BwLIn profile by 7 times. Finally, being divorced, separated or widowed doubled the probability of belonging to the IPB profile. These results are consistent with previous studies that identified various occupational and sociodemographic factors as predictors of burnout in healthcare professionals (e.g., [111–113]). Unexpectedly, being a nursing staff member decreased the probability of belonging to the BwLIn and IPB profiles. This is counterintuitive because nursing staff are particularly vulnerable to stress given their level of empathy [114], which would make them more likely to present burnout. Job satisfaction as a factor that reduces the probability of belonging to a profile with burnout is also consistent with previous studies that identified various personal and psychological aspects of health care personnel as a protective factor against burnout [41-44].

Comments:

Based on your study results, could you provide at least some guesses as to what these interventions could be?

Response:

The section was reworked.

Change:

4.1. Practical Implications of this Work

The practical implications of this work are diverse. Both quantitative and qualitative differences between the two profiles with burnout syndrome (i.e., BwHIn and BwLIn) and the profile that is in process (i.e., IPB) indicate that each of them requires specific attention according to their particularities. Additionally, in this study it was detected that some of the variables that predict belonging to one or another profile are difficult to manipulate (e.g., BwHIn: shift; BwLIn: nurse) because they are constrained by socioeconomic factors or refer to working conditions. present in the care institution were healthcare professionals working. The above implies that the strategies designed should focus rather on psychological variables that allow healthcare professionals to functionally face the demands to which they are exposed. In this regard, there is evidence that moral resilience [41], resilience and social support [43], and self-compassion [44] are factors that prevent deterioration in the mental health of this population, including the presence of Burnout syndrome. However, the psychological variables that affect each of these two profiles and the way in which they do so remain unknown. It is likely that they act in a differentiated manner, which would suggest the need for intervention strategies specific to each profile, instead of generic interventions. The identification of the IPB profile also implies the need for focused attention to its needs, which prevents its members from evolving to either the BwHIn or BwLIn profile. Finally, further investigation of the HeLb profile may be useful to obtain information about the psychological characteristics of healthcare professionals who, although exposed to the same demands as their colleagues with burnout, do not develop this syndrome. The latter could be used in burnout syndrome prevention programs that instruct or enhance these characteristics.

 An unexpected aspect that could have practical implications in burnout research is the fact that divorced, separated or widowed people, although they present burnout, are not found in the BwHIn profile. One possibility is that, by not having a partner, they avoid experiencing marital burnout [Nejatian et al. 2021], which can serve as an additional factor to the work demands to which they are exposed and that can enhance the development of burnout. Future studies should investigate the role of marital burnout in healthcare professionals in order to identify its role in their work burnout.

Reviewer 2 Report

I would encourage the authors to add specific recommendations in the section on Practical implications, and suggestions for further research in their conclusion.  Otherwise, this is a fine paper, worth publishing.  See details in the attached report.

Author Response

Review Report (Reviewer 2)

Comments:

First, after a very explicit description of the four profiles identified by LPA, one would expect somewhat more specific recommendations in the section on the practical implications of this research.

Response:

The section was reworked.

Change:

4.1. Practical Implications of this Work

The practical implications of this work are diverse. Both quantitative and qualitative differences between the two profiles with burnout syndrome (i.e., BwHIn and BwLIn) and the profile that is in process (i.e., IPB) indicate that each of them requires specific attention according to their particularities. Additionally, in this study it was detected that some of the variables that predict belonging to one or another profile are difficult to manipulate (e.g., BwHIn: shift; BwLIn: nurse) because they are constrained by socioeconomic factors or refer to working conditions. present in the care institution were healthcare professionals working. The above implies that the strategies designed should focus rather on psychological variables that allow healthcare professionals to functionally face the demands to which they are exposed. In this regard, there is evidence that moral resilience [41], resilience and social support [43], and self-compassion [44] are factors that prevent deterioration in the mental health of this population, including the presence of Burnout syndrome. However, the psychological variables that affect each of these two profiles and the way in which they do so remain unknown. It is likely that they act in a differentiated manner, which would suggest the need for intervention strategies specific to each profile, instead of generic interventions. The identification of the IPB profile also implies the need for focused attention to its needs, which prevents its members from evolving to either the BwHIn or BwLIn profile. Finally, further investigation of the HeLb profile may be useful to obtain information about the psychological characteristics of healthcare professionals who, although exposed to the same demands as their colleagues with burnout, do not develop this syndrome. The latter could be used in burnout syndrome prevention programs that instruct or enhance these characteristics.

 An unexpected aspect that could have practical implications in burnout research is the fact that divorced, separated or widowed people, although they present burnout, are not found in the BwHIn profile. One possibility is that, by not having a partner, they avoid experiencing marital burnout [Nejatian et al. 2021], which can serve as an additional factor to the work demands to which they are exposed and that can enhance the development of burnout. Future studies should investigate the role of marital burnout in healthcare professionals in order to identify its role in their work burnout.

Comments:

Second, the conclusion, as is often the case, is a bit too short. It should be expanded to include, for example, suggestions for further research.

Response:

The change is made.

Change:

5. Conclusions

Through LPA, two profiles were identified in healthcare professionals that combined burnout syndrome and low work enthusiasm in a specific and mutually exclusive way. A third profile exhibited an inverse pattern, with a low level of burnout and high work engagement. The last profile included people who tended to develop burnout syndrome and low work engagement. The differences between the two profiles with burnout suggest the participation of empathy and compassion fatigue, possibly as mediating variables between burnout and work engagement. This relationship should be investigated in future studies. It is also necessary to design and implement strategies that reverse the tendency to burnout and low work enthusiasm detected as a specific profile. A second job in a government health care institution; a shift other than the morning shift; being divorced, separated or widowed; and the work load are predictors of burnout syndrome and low work engagement. Positive psychological variables, as well-being, that prevent burnout should be investigated.

Reviewer 3 Report

The paper is well written, but it needs some minor expalnation, to make it clearer to the reader

Please explain the Eagly and Chaiken model  and how and if Gil-Monte modified it (p.2);

In which way the Spanish Burnout Inventory is different from the original instrument ?

Please explain the SBI theoretical model distinction between two types of burnout: guilt and nonguilt

Author Response

Review Report (Reviewer 3)

Comments:

Please explain the Eagly and Chaiken model  and how and if Gil-Monte modified it (p.2);

Response:

The attitudinal model of the Eagly and Chaiken was briefly explained.

Change:

From a similar point of view, the Eagly and Chaiken [4] model, that views attitudes as general evaluations in terms of favorability-unfavorability, influenced by affective experiences with the attitudinal object and cognitive components, was used by Gil-Monte [5] to explain the appearance of attitudes and characterized burnout as a response represented by cognitive deterioration (loss of enthusiasm toward the job) and emotional deterioration (psychological exhaustion) with the presence of negative attitudes (indolence), characterized by indifference or cold, distant and sometimes harmful behaviors toward the people to whom a service is provided. Gil-Monte [5] also observed that sometimes negative attitudes are accompanied by feelings of guilt, which represents a key variable in this model and can explain the relationship between burnout and variables such as depression [6].

Comments:

In which way the Spanish Burnout Inventory is different from the original instrument ?

Response:

In the manual of the original instrument Gil Monte points out that the correct way to name the instrument in English is Spanish Burnout Inventory, so these are not two different instruments, but rather it is the correct way to name it in English.

Change:

No changes were made because it is the same instrument.

Comments:

Please explain the SBI theoretical model distinction between two types of burnout: guilt and nonguilt.

Response:

The distinction between a burnout profile with guilt and one without guilt was briefly specified.

Change:

The SBI [81,82] consists of 20 items answered on a Likert scale with 5 response options (i.e., 0: never to 4: very frequently or every day) organized into 4 dimensions: enthusiasm toward the job, psychological exhaustion, indolence, and guilt. The score for each dimension is determined by the arithmetic sum of the scores for each item. The total score is calculated as the sum of the dimensions enthusiasm toward the job, psychological exhaustion and indolence divided by 15. The correction of this last score allows for its transformation into percentiles, and the burnout level was interpreted as very low (<11), low (11-33), medium (34-66), high (67-89), or critical (>89) [18]. The SBI theoretical model distinguishes between two types of burnout: guilt and nonguilt, whose difference lies in the presence or absence of guilt when detecting negative attitudes towards work. Nonguilt is defined as scores in the 4 dimensions equal to or greater than the 90th percentile. This instrument has been validated in Spanish with a population of various nationalities, including Mexican [18]. The psychometric properties of the confirmatory model with 4 factors were excellent (x2/df = 1.82; CFI = 0.99; TLI = 0.99; RMSEA = 0.04 (90% CI = 0.03 to 0.05)) except for x2 (298.508, p < 0.01) and WRMR (1.05). The internal consistency for each dimension was α = 0.72, 0.87, 0.77, and 0.82 for enthusiasm toward the job, psychological exhaustion, indolence, and guilt, respectively.